# Insights into the Taxonomically Challenging Hexaploid Alpine Shrub Willows of *Salix* Sections *Phylicifoliae* and *Nigricantes* (Salicaceae)

**DOI:** 10.3390/plants12051144

**Published:** 2023-03-02

**Authors:** Natascha D. Wagner, Pia Marinček, Loïc Pittet, Elvira Hörandl

**Affiliations:** Department of Systematics, Biodiversity and Evolution of Plants (with Herbarium), University of Goettingen, Untere Karspüle 2, D-37073 Göttingen, Germany

**Keywords:** biogeography, genomics, morphometrics, polyploidy, taxonomy

## Abstract

The complex genomic composition of allopolyploid plants leads to morphologically diverse species. The traditional taxonomical treatment of the medium-sized, hexaploid shrub willows distributed in the Alps is difficult based on their variable morphological characters. In this study, RAD sequencing data, infrared-spectroscopy, and morphometric data are used to analyze the phylogenetic relationships of the hexaploid species of the sections *Nigricantes* and *Phylicifoliae* in a phylogenetic framework of 45 Eurasian *Salix* species. Both sections comprise local endemics as well as widespread species. Based on the molecular data, the described morphological species appeared as monophyletic lineages (except for *S. phylicifolia* s.str. and *S. bicolor*, which are intermingled). Both sections *Phylicifoliae* and *Nigricantes* are polyphyletic. Infrared-spectroscopy mostly confirmed the differentiation of hexaploid alpine species. The morphometric data confirmed the molecular results and supported the inclusion of *S. bicolor* into *S. phylicifolia* s.l., whereas the alpine endemic *S. hegetschweileri* is distinct and closely related to species of the section *Nigricantes*. The genomic structure and co-ancestry analyses of the hexaploid species revealed a geographical pattern for widespread *S. myrsinifolia*, separating the Scandinavian from the alpine populations. The newly described *S. kaptarae* is tetraploid and is grouped within *S. cinerea*. Our data reveal that both sections *Phylicifoliae* and *Nigricantes* need to be redefined.

## 1. Introduction

Polyploidy occurs across the tree of life [1,2,3]. The presence of multiple gene copies in polyploids allows for gene neo- and subfunctionalizations, epigenetic changes, and consequently a differential expression of homeologous genes [4,5]. Additionally, polyploidy provides larger physiological and phenotypic flexibility to respond to different environmental conditions [3,6,7,8], which facilitates the colonization of various ecosystems [9,10,11].

Especially allopolyploidization (formed by hybridization between different species/lineages followed by genome duplication) is considered particularly likely to create biotypes with novel genomic features [2,12,13]. Polyploidy creates an immediate reproductive barrier against the diploid progenitors [12,14] and sexual polyploids can form distinct evolutionary lineages and be recognized as species [15]. However, the gene flow between polyploid lineages is possible especially on the same ploidy level (e.g., [16]). Thus, the evolutionary origins of polyploid clades can be complex and involve multiple origins, more than two progenitors, but also hybridization between progenitors before polyploidization (e.g., [17]). For genera with polyploid clades, the subgeneric classification has been notoriously difficult, because of the reticulate, non-hierarchical relationships of groups (e.g., [18,19]).

The genus *Salix* L. (Salicaceae) comprises about 450 species of dioecious trees and shrubs mainly occurring in the Northern Hemisphere [18,20]. About 40% of species are polyploids [21]. Willows are important elements of various kinds of natural wetlands, riparian vegetation, and arctic–alpine tundras, and are involved in many biotic interactions (e.g., [22,23,24]). The *Chamaetia*/*Vetrix* clade comprises about three quarters of the described species diversity in the genus *Salix*, containing more than 350 species classified in about 40 sections. Some species of this clade are adapted to cold, hostile environments of the arctic–alpine zone and show a decumbent or creeping growth (“dwarf shrubs”), while others are medium-sized shrubs and small trees. The European Alps harbor 33 willow species, of which 28 belong to the *Chamaetia*/*Vetrix* clade (so-called “shrub willows” [25]), showing that willows are an important part of alpine plant diversity. However, the determination of alpine shrub willows is difficult because of dioecy, simple, reduced flowers, common natural formation of hybrids, and large intraspecific phenotypic variation [20,24,26]. Especially the latter point led to the description of many species, and the overall taxonomy of *Salix* is still far from being resolved [27]. The presence of polyploidy further increases these difficulties. Especially the described hexaploid species tend to be morphologically diverse [24,25], although taxonomically accepted [20,24,28]. However, the phylogenetic relationships of the medium-sized, hexaploid shrub willows of the sections *Nigricantes* and *Phylicifoliae* have never been tested using a comprehensive sampling.

Recent molecular studies based on RAD sequencing data were able to produce robust phylogenetic trees, resolving the relationships of diploid and tetraploid Eurasian shrub willows [29,30]. The included alpine species did form monophyletic lineages in these molecular studies, which show that SNP-based methods based on RAD loci were suitable to analyze polyploid *Salix* species [30]. In this paper, we apply this technique to conduct molecular analyses on the two sections containing hexaploids. However, there are limitations to such methods, especially regarding the simplifications of SNP calling in a consensus sequence that might blur some of the results for the higher polyploids. Therefore, we combine the data with non-molecular approaches. Infrared spectroscopy (sometimes also called “NIRS” (near-infrared spectroscopy)) of leaves is emerging as a promising tool to analyze plant phylogenetic diversity as well as certain chemical compounds [31,32,33]. This method uses the structural and chemical traits of leaves assessed by specific reflectance patterns to differentiate groups at the species or population level with a high accuracy. Leaf spectroscopy is non-destructive, fast, cheap, and reliable [34], and thus can be easily applied to dried leaf material (herbarium material). Additionally, it can be applied to any ploidy level.

Previous taxonomic treatments of species in willows have been exclusively based on descriptive morphology. Morphological characters, especially of vegetative parts, are highly variable in willows due to high phenotypic plasticity and gradual changes during development in the course of one season [20,35]. Therefore, even leaves of the same individual exhibit vast variation of diagnostic characters. Morphological studies resulted in several different taxonomic circumscriptions, specifically in the species complex of the European *Phylicifoliae* group sensu Rechinger [28] (including *S. phylicifolia* L., *S. bicolor* Ehrh. ex Willd., and *S. hegetschweileri* Heer) by different authors (Table 1). The three taxa were diagnosed only by a few leaf characters (shape, margin, stipules, and indumentum). To shed light on questionable species circumscriptions, comparative morphometric studies can help to quantify objectively the similarity/dissimilarity of taxa and estimate the taxonomic value of diagnostic traits [36]. Therefore, we apply a morphological analysis on the target species in addition to molecular data and infrared spectroscopy.

In this paper, we use a comprehensive dataset to test the taxonomic classification and to analyze the phylogenetic relationships of several taxonomically challenging polyploid species of the section *Nigricantes* sensu Skvortsov [20] and section *Phylicifoliae* sensu Rechinger [28]. The first comprises *S. myrsinifolia* (6x), widespread in Eurasia, *S. mielichhoferi* (6x), endemic to the Eastern Alps, *S. apennina* (6x) from the Apennines, *S. cantabrica* (4x) from Spain, and the recently described *S. kaptarae* endemic to Crete (Figure 1). For section *Phylicifoliae* (in Eurasia subsect. *Bicolores* in [20] and *S. phylicifolia* group sensu Rechinger [28]; see Table 1), we consider widespread *S. phylicifolia* L. (6x) and *S. bicolor* Ehrh. ex Willd. (3x, 6x) with a scattered distribution in Europe as well as the tetra- to hexaploid *S. hegetschweileri* Heer em. Buser, an alpine endemic species (Figure 1). The two sections are morphologically similar, and previous authors hypothesized close relationships between the groups [18,20,28]. We include additional species of both sections and combine the molecular data with the already published RAD sequencing data for European *Salix* species to analyze the position of the species in a phylogenetic framework. Subsequently, we apply genetic structure and co-ancestry analyses to obtain insights into the genomic composition of the hexaploids. In addition, we test whether leaf reflectance spectra bear phylogenetic signals within the hexaploid species and provide further information on taxonomic classification. Finally, we test species delimitation within the *Phylicifoliae* group with a morphometric analysis. By including specimens from the type locations of *S. bicolor* and *S. hegetschweileri* into both molecular, infrared spectroscopic and morphometric analyses (see Materials and Methods, Appendix A), we can fix the nomenclature for final taxonomic conclusions.

## 2. Results

### 2.1. Phylogenetic Analyses and Genetic Structure Based on RAD Sequencing Data

RAD sequencing yielded an average of 7.26 Mio reads per sample. We used up to five accessions per species for the analyses of the backbone topology to reveal the position of members of the sections of interest in the phylogeny without obscuring the results too much by mixing several ploidy levels. The ipyrad pipeline of 145 samples representing 45 species revealed 46,568 RAD sequencing loci containing 630,764 SNPs. The concatenated sequence alignment comprised 66.95% of the missing data. The results of the RAxML analyses are presented in Figure 2, and the same dataset including QS scores is shown in Appendix A. The species *S. triandra* was used as the outgroup to root the tree. Three major clades could be observed, while *S. reticulata* was in a sister position to all remaining samples. The hexaploid species *S. myrsinifolia, S. mielichhoferi, S. apennina*, and *S. hegetschweileri* form a clade that showed slightly discordant and skewed QS values and medium-supported BS support. It occurred in a sister position to a clade comprising the species *S. caprea*, *S. aurita*, *S. appendiculata*, *S. cinerea*, *S. atrocinerea*, *S. laggeri*, and *S. salviifolia*. *Salix cantabrica*, taxonomically assigned to the section *Phylicifoliae,* appeared in a sister position to the just mentioned clade. *Salix kaptarae*, a recently described species from Crete and assigned to the section *Nigricantes*, occurred within the widespread lowland species *S. cinerea*.

The Fennoscandian hexaploid species *S. phylicifolia* and triploid/hexaploid *S. bicolor* did not form clades according to their species identity but appeared intermingled within a paraphyletic group. Each subclade showed only low QS and BS support. However, accessions of both species were situated in a sister position to a clade containing two subclades, one including *S. lapponum*, *S. helvetica, S. foetida,* and *S. waldsteiniana* and a subclade containing the species *S. viminalis*, *S. schwerinii*, and *S. rehderiana* (Figure 2 and Appendix A). The remaining topology is in accordance with previously published data on the *Salix* subg. *Chamaetia*/*Vetrix* clade [25,30].

Subsequently, a RAxML analysis, including all collected samples of the analyzed sections as well as a reduced backbone sampling, was conducted to analyze the tree topology of both sections in more detail. The results are shown in Appendix A. The samples of *S. myrsinifolia*, *S. hegetschweileri*, *S. apennina*, and *S. mielichhoferi* form one clade. Within this clade, *S. myrsinifolia* was separated into two subclades: One containing samples from Scandinavia as well as Russian *S.* cf. *jenisseensis*. The other clade contained samples from the Alps. Samples of *S. mielichhoferi* formed a monophyletic group with a strong support. The accessions of *S. apennina* were grouped in a sister position to *S. mielichhoferi*. *Salix hegetschweileri* is not monophyletic but forms two clades as a paraphyletic group in a basal position to the remaining species. The accessions of *S. bicolor* also form two clades, one clade including samples from Switzerland and the other samples from Germany and Austria in a sister position to samples of *S. phylicifolia.*

The results of the co-ancestry analysis of the species belonging to the section *Nigricantes* and *Phylicifoliae* are shown in Figure 3. The analysis was based on 43 samples and 49,754 RAD loci. *Salix phylicifolia* s.l., including *S. phylicifolia* and *S. bicolor,* was clearly separated from the remaining species. *Salix hegetschweileri* shared co-ancestry with species of the section *Nigricantes*, however, the analysis also revealed some co-ancestry with the *Phylicifoliae* group indicated by the orange color. *Salix myrsinifolia* was divided into two groups. One includes all samples from the European Alps. The other group comprises samples originating in Fennoscandia and Estonia as well as *S.* cf. *jenisseensis* from Russia. The latter clade showed some shared ancestry with Nordic *S. phylicifolia*.

To analyze the two sections containing the alpine hexaploid species in more detail, we conducted a separate analysis for the two observed major clades of each section *Nigricantes* plus *S. hegetschweileri,* and *Phylicifoliae* (Appendix A). For the section *Nigricantes*, 29 samples of *S. mielichhoferi*, *S. myrsinifolia*, *S. apennina*, and *S. hegetschweileri* were included. The ipyrad pipeline obtained 8,164 RAD loci showing 90,756 SNPs and 15.7% missing data on a restrictive filtering. For twelve samples of *S. phylicifolia* and *S. bicolor* (*Phylicifoliae*), 27,010 RAD loci (149,614 SNPs, 6.51% missing data) were observed with the ipyrad pipeline. The results of the co-ancestry analysis with fineRADstructure for the sections *Nigricantes* and *Phylicifoliae* are illustrated in Appendix A. The co-ancestry analysis based on 8149 shared loci revealed separate clusters for each species included. Overall, however, the red–orange colors showed moderate levels of co-ancestry within all hexaploids included in this paper. Shared loci were observed for *S. mielichhoferi* and *S. myrsinifolia*. Additionally, shared loci were present for *S. hegetschweileri* with *S. myrsinifolia*. For the subclade of the section *Phylicifoliae*, the samples from Harz Mountain were separated and showed very low levels of co-ancestry (yellow color). Triploid samples from Austria showed only low levels of co-ancestry with the remaining samples, while hexaploid samples of *S. phylicifolia* and *S. bicolor* from Switzerland showed a high number of shared loci indicated by orange and red colors.

The results of the genetic structure analyses with sNMF for all samples and both clades observed in the Maximum Likelihood tree are displayed in Appendix A.

The overall analysis revealed some admixture in *S. hegetschweileri* and *S. jenisseensis c.f.* for k = 2, while the other individuals share genetic partitions according to their sectional classification (see Appendix A). The overall as well as the clade-wise results indicate that the individuals of *S. bicolor* and *S. phylicifolia* do not form two distinct groups according to their taxonomic species identity. Instead, two samples of *S. bicolor* from Harz Mountain share the same structure (light blue), while all remaining accessions show a different genomic composition (dark blue) independent of species determination (k = 2). A population-wise structure is present for k = 6 (Appendix A).

For the *Nigricantes* clade, the sNMF results for k = 6 are presented in Appendix A. *Salix mielichhoferi* and the majority of samples of *S. myrsinifolia* share a genetic cluster (yellow), while *S. myrsinifolia* shows a certain amount of admixture. Two samples of *S. mielichhoferi* show a distinct cluster (light blue). Samples of *S. myrsinifolia* originating from Fennoscandia as well as *S.* cf. *jenisseensis* form a distinct cluster (orange). *Salix apennina* and *S. hegetschweileri* share a genetic cluster (dark blue).

### 2.2. Infrared Spectroscopy Data

The results of the infrared spectroscopy data of all hexaploid species are shown in Figure 4. *Salix phylicifolia* and *S. bicolor* are distinct from *S. myrsinifolia*, which showed a variable spectrum, and *S. mielichhoferi*. The two included samples of *S. hegetschweileri* (after filtering) were bridging *S. myrsinifolia* and *S. phylicifolia*. The prediction table showed high prediction values for *S. mielichhoferi* and for *S. bicolor* and *S. phylicifolia* when combining the latter two into *S. phylicifolia* s.l.. Most *S. myrsinifolia* samples were assigned to *S. mielichhoferi*. The two measurements of *S. hegetschweileri* were assigned to *S. phylicifolia* s.l. The more detailed results of the infrared spectroscopy of leaf material for the section *Nigricantes* (*S. mielichhoferi*, *S. myrsinifolia*, and *S.* cf. *jenessiensis*) as well as for the section *Phylicifoliae* (*S. bicolor*, *S. hegetschweileri*, and *S. phylicifolia*) are illustrated in Appendix A. The results shown as PCA and density plots show discrimination between the species *S. mielichhoferi* and *S. myrsinifolia* based on their filtered spectra. However, the measurement of *S.* cf. *jenessiensis* falls into *S. myrsinifolia*. For the section *Phylicifoliae*, *S. bicolor* and *S. phylicifolia* are in close proximity based on the PCA. The first coordinate explains 54.1% of the variation in the dataset. When including information on ploidy (3n and 6n), the results show a similar, but slightly more distinct pattern.

### 2.3. Ploidy Level Determinations for S. bicolor and S. kaptarae

The FC results (Appendix A) show a hexaploid genome size for *S. bicolor* samples from Switzerland as well as for the two samples from the Harz Mountains, Germany, which represent the only remaining population from the locus classicus. *Salix kaptarae* showed a tetraploid genome size.

### 2.4. Morphometric Data of S. phylicifolia, S. bicolor and S. hegetschweileri

The characters of leaf length, leaf width, angle of the blade base, and length of stipules showed significant differences (*p* > 0.05) between *S. hegetschweileri* and both *S. bicolor*/*S. phylicifolia*, whereas the two latter species did not differ from each other (Figure 5). *Salix hegetschweileri* has longer and wider leaves, a broader angle of the blade base, and more pronounced stipules compared to the other taxa. Length/width ratios of leaves, however, did not show any differences between taxa (Figure 5c). Length of teeth differed in all three taxa among each other, with *S. bicolor* having entire margins or very short teeth, followed by tiny teeth in *S. phylicifolia* and pronounced teeth in *S. hegetschweileri* (Figure 5f).

PCA using five characters and plotted with convex hulls for the three taxa suggested strong overlaps of *S. bicolor* and S. *phylicifolia* (Appendix A), whereas PCA with convex hulls for only two taxa according to genetic data (*bicolor* + *phylicifolia* and *hegetschweileri*) revealed only a slight overlap of the two clusters (Figure 5g). Leaf length and angle of the base have the strongest correlations (highest loading values) with the first two axes. PCoA with convex hulls for two taxa (*bicolor* + *phylicifolia and hegetschweileri*) revealed two clearly distinct clusters in the scatter plot (Figure 5h), whereas *S. phylicifolia* and *S. bicolor* visualized separately did not show a dissimilarity gap. The discriminant analysis with three predefined taxa revealed a confusion matrix with 92.5% of samples correctly classified, whereas with two predefined groups (*S. bicolor* + *phylicifolia* and *S. hegetschweileri* as in Figure 5g,h) revealed 97.5% of samples correctly classified.

## 3. Discussion

The complexity of the allopolyploid genome, especially for higher polyploids as studied in this paper, is a source of variable phenotypic and physiological traits [3,7]. However, although hexaploid *Salix* species are morphologically variable and thus taxonomically challenging, the reduced representation data of genomes analyzed with bioinformatic tools were able to resolve the relationships of the hexaploid species analyzed in this paper. Our study showed that spectral data can be a useful supplement to analyze phylogenetic relationships, which is in line with other studies [31,32,33]. In combination with molecular and morphometric data, we were able to answer questions on the taxonomic treatment of alpine hexaploid *Salix* species. Despite the mosaic-like genomic composition, our results show clear species delimitation for all species, except *S. phylicifolia*/*bicolor*. This might be due to successful sexual reproduction and the establishment of the polyploid lineages as observed in tetraploid willows [30]. Nevertheless, our data indicate introgression and geneflow between the hexaploid genomes (e.g., Figure 3). In the following, we discuss our methodical approach, the evolutionary history, and the impact of our results on the taxonomic treatment of the two sections *Nigricantes* and *Phylicifoliae*.

### 3.1. Comparison of Datasets

In this study, we used RAD loci and SNPs to analyze the phylogenomic relationships in hexaploid species. RAD sequencing is a reduced representation method and, because we used a de novo assembly, we generated short anonymous loci. However, the advantage of this technique is the considerable number of generated SNPs that could be successfully used for species delimitation in hexaploids. The reduction in the complexity of up to six alleles into one single (ambiguous) consensus sequence allowed us to combine samples with different ploidy levels. To find the best balance between missing data and number of informative SNPs, we tested different parameter sets. The high amount of missing data (~70%) in the final dataset used for the backbone topology did not affect the phylogenetic reconstruction. This is in accordance with [38]. However, to display (allo)polyploid species in a bifurcating tree is not the best way to address a network-like evolutionary pattern [2]. Thus, we implemented genetic structure analyses with sNMF and co-ancestry analyses with fineRADstructure. Both tools can handle different ploidy levels [39,40]. The close relationships of species within subclades allowed for more conserved settings with less locus dropout and, thus, less missing data in the final datasets. Our study is one of the few available studies applying these two tools to hexaploid non-model plant species. We showed that these population genomic tools are suitable for (polyploid) species complexes and/or closely related species to reveal the genomic structure and co-ancestry, which is in accordance with other studies [41,42,43]. However, we realized that genetically (almost) identical individuals (=clones) from the same populations blurred the results of both tools and tended to form separate clusters (see, e.g., *S. bicolor* from Harz Mountain or *S. mielichhoferi* from Austria). In these cases, the signal is stronger than the co-ancestry with ally species. Similar patterns were observed in a study on an alpine hybrid zone of two willow species [44]. We, therefore, recommend excluding clonal or highly similar individuals from this sort of analyses.

Subsequent to the molecular dataset, we applied infrared spectroscopy to the target species, which is not sensitive to ploidy. This technique (also called “near-infrared spectroscopy (NIRS)”) became a popular tool in forestry in the 2000s, where leaf spectra were used for tree species discrimination and for (the prediction of) foliage chemical compounds in the field (reviewed in [34]). The main reasons for the frequent use of spectroscopy are its low cost as well as the non-destructive, fast, and reliable measurements. The utility of spectral data for taxonomic purposes has been demonstrated in several plant species [45,46]. Stasinski et al. [31] used spectral data in combination with molecular data (genotyping-by-sequencing, GBS) for a fine-scale diversity analysis in two hybridizing arctic shrub species of genus *Dryas*. The authors demonstrated that reflectance spectroscopy captured genetic information that can be used to accurately classify leaves of species, hybrids, and populations in a taxonomically challenged group. Our data showed good differentiation between the two analyzed species groups (Figure 4) but only low differentiation within species. However, our dataset comprised much fewer measurements per species than in the studies mentioned above and we experienced a bias in the prediction results towards the species with a higher number of measurements (e.g., *S. myrsinifolia* samples were predicted as *S. mielichhoferi*, and *S. hegetschweileri* as *S. phylicifolia*, while we measured in both cases more samples for the latter one). Additional samples might stabilize the prediction in the PLS-DA. The combination of spectral data and morphological data in the genus *Myrcia* (Myrtaceae) by Gaem et al. [47] demonstrated that the multidimensional natures of the entire spectra were very efficient in assigning individuals to species categories. Our results support the ability of this technique to discriminate different *Salix* species within one section. Even better, we are able to discriminate different species groups and, thus, our results are in line with the above-mentioned studies. Additionally, we observed a considerable within-species variability in *S. myrsinifolia* and *S. bicolor*. However, we did not consider collection time (leaf age) and collection site (ecological factors), which might affect the chemical composition of the leaves of these highly variable plant species. The leaves of *S. myrsinifolia* and *S. mielichhoferi* turn black when dried and this might further contribute to the observed diversity of measurements (see Figure 4a). Pigments absorb light in the visible region (400–700 nm), whereas light in the near infrared region (700–1100 nm) is scattered by leaf anatomical, tissue, water, and surface features, and light in the short-wave infrared region (1400–2500 nm) is scattered and absorbed by anatomical features and biochemicals, such as cellulose, phenolics, and water [31]. Thus, leaf spectra are complex datasets influenced by both environmental and genetic factors. Overall, considering the fast and easy use of this technique, it is a valuable supplement to molecular and morphological datasets.

Finally, we addressed the delimitation of species of the previous section *Phylicifoliae* with a morphological analysis, which confirmed *S. hegetschweileri* being distinct from *S. phylicifolia*/*S. bicolor*. At the species and individual levels, we can confirm a high variability of characters that is typical for willows [20,48]. A certain overlap of phenotypic variation of genetically clearly distinct species was also observed in other molecular–morphometric studies on willows [49,50]. In addition to gradual changes during seasonal development, phenotypic plasticity depending on habitat conditions causes large individual variation. For instance, leaf size usually decreases on dry soils, which applies to our dataset, specifically to *S. hegetschweileri* individuals found on the rather dry glacier moraines of the Gletschboden population. Only these small-leaved individuals caused the overlap of variation with the *S. bicolor*/*S. phylicifolia* cluster in the displayed PCA (Figure 5g). However, morphometric data in combination with molecular data are powerful to separate closely related taxa, as it has been demonstrated in many other polyploid complexes [51,52,53].

### 3.2. Evolutionary History and Biogeography

Previous studies on polyploid willows suggested allopolyploid origins for most polyploid species [30]. Skvortsov [20] already emphasized that willow species are characterized by distinct eco-geographical distributions. In the *Nigricantes* clade, it is plausible that all polyploids speciated via geographical and ecological isolation. *Salix apennina*, *S. hegetschweileri*, and *S. mielichhoferi* have allopatric distributions. Only in the Alps, *S. myrsinifolia* is sympatric with *S. hegetschweileri* in the western Central Alps and with *S. mielichhoferi* in the Eastern Central Alps, but in both areas, the two latter species occur in subalpine shrubberies, whereas *S. myrsinifolia* occurs mostly in the montane forest zone in wet meadows and along rivulets [37,54]. The alpine endemic species *S. mielichhoferi* seems to have evolved as a separate evolutionary lineage out of the Central European populations of widespread *S. myrsinifolia* via ecological speciation. It could have originated after the last glacial maximum, since the current distribution area of *S. mielichhoferi* was mostly covered by ice at the last glacial maximum (LGM), except for the southernmost parts in Northern Italy [37,55]. Alternatively, the species predated the LGM and survived in the southern peripheral calcicolous refugia between the Lake Como and the Dolomites [55]. Both species share a certain amount of their genomes (Figure 3 and Appendix A) but are clearly separated into two lineages (Figure 2 and Appendix A). However, the shared genomic compartments might also be explained by recent gene flows, since they overlap in their elevational distribution in the Alps in the subalpine zone, where intermediate forms between both species were observed in the contact zone [54]. The hybrid origin of such individuals, however, needs to be confirmed by further studies.

The evolutionary origin of *S. hegetschweileri* has long been under dispute, also due to unclear delimitation against *S. bicolor* and *S. phylicifolia* and the notorious confusion of taxa (Table 1; reviewed in [54]). Based on our data, we considered *S. hegetschweileri* a distinct lineage and alpine endemic species (see Section 3.4). The survival of glacial maxima could have been possible in southwestern peripheral refugia, where many siliceous bedrock conditions are available [55]. This silicolous species is very successful as a colonizer of glacier forefields and could have been one of the first postglacial pioneers of the alpine flora after glacier retreat.

The evolution of the *S. phylicifolia*–*bicolor* alliance can be best explained by geographical isolation. During the LGM (or during previous cold periods), *S. phylicifolia* s.l. probably colonized the whole ice-free tundra zone between the big ice shields covering the Alps and Northern Europe. During the postglacial recolonization, *S. phylicifolia* migrated mainly northwards and left some isolated relic populations in the Jura and adjacent western Alps, in the Harz Mountains, the Riesengebirge (not included in this paper), and in the Eastern Alps in a refugial area on siliceous bedrock (see [55]). The disjunct clades of the Swiss and Austrian populations of “*S. bicolor*” might be due to geographic isolation during LGM. Postglacial reforestation hindered range expansion and gene flow between the isolated relic Central European willow populations. The population structure observed in the co-ancestry analyses could be explained by the disjunct distribution of these populations classified as “*S. bicolor*”.

### 3.3. Taxonomy of the Section Nigricantes

Most taxonomically described species within this section can be confirmed by our results. The infrared spectroscopy data were able to distinguish *S. mielichhoferi* and *S. myrsinifolia*, and in the molecular datasets, species of this section were clearly discriminated. The widespread and morphologically diverse *S. myrsinifolia* is divided into two clades, separating the Scandinavian accessions from the alpine accessions (Figure 2 and Figure 3). According to [20], the widespread *S. myrsinifolia* was divided into three subspecies. For Fennoscandia, *S. myrsinifolia* ssp. *borealis* was described. We did not infer subspecific treatment in this paper since our main interest was on the alpine hexaploids. However, the separate clade for Fennoscandian *S. myrsinifolia* could be due to the geographical separation of these samples. The observed low degree of co-ancestry (Figure 3) with Scandinavian *S. phylicifolia* could be due to recent introgression. Whether this group merits a taxonomic treatment as subspecies should be tested with an expanded sampling. *S. myrsinifolia* and *S. mielichhoferi* were accepted by all authors as distinct species. Morphologically, they are differentiated by the characteristics of leaves and twigs [24] and the separation was supported by our results. *Salix apennina* was nested within the Nigricantes clade, and all accessions form a monophyletic group. *Salix hegetschweileri*, which was previously regarded as part of Sect. *Phylicifolia* (see discussion below), is sister to the *S. apennina*–*myrsinifolia*–*mielichhoferi* clade.

*Salix kaptarae*, a recently described species from Crete [56], was described as closely related to *S. apennina* based on morphological characters and taxonomically assigned to section *Nigricantes*. However, our data did neither confirm the proximity to *S. apennina* as suggested by the authors, nor the assignment to the section *Nigricantes*. Instead, our data indicate the inclusion of this species into *S. cinerea* (Figure 2, and Appendix A) within the section *Vetrix* sensu [20]. The morphological characters of *S. kaptarae* such as pubescent branches, leaf morphology, and pronounced striae on decorticated wood (pers. observation of N. Wagner), fit the assignment to this section. Additionally, the sample included is tetraploid based on our flow cytometry data (Appendix A), as it is also documented for *S. cinerea* (*2n* = 76; [57,58,59]). Although the flora of Crete has been well studied for decades, in the vascular plant checklist of Crete only *S. alba* was mentioned [60]. The population of *S. kaptarae* was mentioned in the year 2000 by Jahn in [61] and later by [62]. That leads to the assumption that the individuals of *S. kaptarae* (*S. cinerea*) were introduced to Crete either via long-distance dispersal or, more likely, by humans. However, we included only one sample from the type location in our molecular dataset. More studies will be needed to analyze the spatio-temporal evolution and taxonomic treatment of *S. kaptarae* in more detail.

The previously circumscribed Sect. *Nigricantes* appears to be polyphyletic in phylogenomic analyses and should be refined to the clade including *S. apennina*, *S. myrsinifolia*, *S. mielichhoferi*, and *S. hegetschweileri*. A shared character of this section that differs from the *Phylicifoliae* is the formation of pronounced stipules [20]. The position of putative *S. jenisseensis*, which was classified in Section *Glabrella* by [20], needs to be clarified with more samples.

### 3.4. Taxonomy of the Section Phylicifoliae

Traditionally, this section included *S. phylicifolia*, *S. bicolor*, *S. cantabrica,* and *S. hegetschweileri* along with other species not included in this paper [20]. Based on our results, this section is polyphyletic. The Iberian *S. cantabrica* was not the focus of our study on alpine hexaploids of both sections, so we included only two samples. However, the results suggest that this species is closely related to members of section *Vetrix* (Figure 2).

The morphological analyses of the section *Phylicifoliae* clearly separate *S. hegetschweileri* from *S. phylicifolia* and *S. bicolor* (Figure 5). Less clear are the results of the spectral data that show high similarity between *S. hegetschweileri* and *S. phylicifolia* (Figure 4 and Appendix A). However, only two samples were included, which is not enough to cover the species diversity. Skvortsov [20] regarded *S. hegetschweileri* a geographically separated, Central European subspecies of *S. phylicifolia* (as ssp. *rhaetica*). Lautenschlager [63] proposed that *S. hegetschweileri* is a recent hybrid between “*S. bicolor*” and *S. myrsinifolia*. He crossed *S. hegetschweileri* individuals from the locus classicus (Urserental, Switzerland), which he misidentified as *S. bicolor*, with *S. myrsinifolia* and obtained *S. hegetschweileri*-like offspring, which he then misinterpreted as “*bicolor*” x *myrsinifolia*-hybrids. Our molecular analyses clearly indicate that populations from Urserental, Gletschboden, and Tyrol have nothing to do with “*S. bicolor*” but are *S. hegetschweileri.* Our data do not support the hypotheses of a homoploid hybrid origin out of two hexaploid extant parents. Our phylogenetic data (Figure 2, Appendix A, and Figure 5) instead separate *S. hegetschweileri* from other members of this section. The alpine endemic is a distinct lineage and shows close relationships to the *Nigricantes* clade in phylogenetic analyses. Based on the molecular and morphological data, we assumed the post-origin local introgression of *S. hegetschweileri* with *S. myrsinifolia* in subalpine contact zones, which also sometimes blurs the morphological differences between these two distinct species (Appendix A). However, *S. hegetschweileri* does show a genetic contribution from *S. bicolor*/*phylicifolia* (Figure 3 and Appendix A). Thus, we cannot rule out ancient contributions from the ancestors of the *phylicifolia* lineage nor recent gene flows. Taken together, we support the taxonomical treatment following [28] and [24], treating *S. hegetschweileri* as a distinct species and alpine endemic.

The taxonomic treatment of *S. bicolor* has long been under discussion (Table 1). Our data suggest that “*S. bicolor*” represents disjunct relic populations of former widespread *S. phylicifolia*. The flow cytometry data showed a hexaploid genome size for the populations of the locus classicus from the Harz Mountains as well as from the Jura and adjacent Swiss Alps (Appendix A). The triploid population from Austria seems to be a single isolated clone of female plants [54] that either evolved from a hexaploid parent via haploid parthenogenesis (meiosis complete, but embryogenesis without fertilization of reduced egg cells), which can happen occasionally in many angiosperms (e.g., [64]). Alternatively, the triploids represent a population that originated from a diploid parent, underwent polyploidization, and became stuck in a triploid bridge stage. A “triploid bridge” is usually formed in the process of polyploidization via unreduced gametes, but in dioecious plants, this process could severely disturb the balance of proportions of sex chromosomes, and hence proportions of male and female individuals [65]. We regard the first scenario as more likely as all other *S. bicolor* samples from the western Alps, Jura, and Harz Mountains were hexaploid. In any case, the triploids represent a special population. Our morphological analysis also includes *S. bicolor* into *S. phylicifolia* (Figure 5). The discrimination of both species in the spectral data analyses as well as the molecular data is rather weak (Figure 2 and Appendix A).

*Salix phylicifolia* s.l. (incl. *S. bicolor*) is, in our present phylogeny, a paraphyletic group (Figure 2). Paraphyly is a frequent phenomenon in polyploid species and can be referred to progenitor-derivative relationships within a dataset [66,67]. Paraphyletic groups are based on shared ancestry and, hence, can be used for species classification [66,67]. The section *Phylicifolia*, however, is polyphyletic, and needs a new circumscription, which requires further studies, including other Eurasian and North American species [18] that we did not sample here.

## 4. Materials and Methods

### 4.1. Studied Material

We sampled species of the sections *Nicricantes* A. Kern and *Phylicifoliae* (Fries) Andersson with specific emphasis on the alpine species of both sections. *Salix myrsinifolia* Salisb., which is hexaploid, widely distributed in Europe, and morphologically highly diverse, has a long tradition of taxonomic discussion. The leaves of this shrub or small tree turn black when they are dried. *S. myrsinifolia* is probably the most variable species of all the European willows [20,24,54]. This species hybridizes with other willows, e.g., *S. glabra*, *S. mielichhoferi*, and *S. cinerea*; however, most plants that have been treated as hybrids are, according [20], “variants within the species variability range”. Several species names were synonymized to *S. myrsinifolia* (e.g., *S. nigricans* Sm. is a younger synonym). Two subspecies are currently accepted (by the authors of Table 1): *S. myrsinifolia* ssp. *myrsinifolia*, and ssp. *borealis* (Flod.) A. Skvortsov. In this study, we sampled *S. myrsinifolia* originating from the Alps and from Fennoscandia. One of our samples matches *S. jenisseensis* (F.Schmidt) Flod. from Russia, a species related to *S. myrsinifolia.* Due to the lack of samples, the identification is provisional.

*S. mielichhoferi* Saut. is a hexaploid species [57,59] that occurs in Austria and Northern Italy and is endemic to the Alps, and also belongs to the section *Nigricantes*. Morphologically, it differs from *S. myrsinifolia* by glabrous, more oblong leaves without a pruinose layer on the lower surface. Both species share the blackening of leaves when dried. However, hybridization with *S. myrsinifolia* is possible [54], leading to intermediate forms.

Additionally, the accessions of *S. apennina* A. K. Skvortsov, a hexaploid species from the Apennines, and one sample from the locus classicus of *S. kaptarae* Cambr.Brullo&Brullo were included. The latter species was recently described from Crete [56] and assigned to the section *Nigricantes.* Based on the morphological characters, it was assumed as being related to *S. apennina*.

The species *S. phylicifolia* L., *S. bicolor* Ehrh. ex Willd. as well as tetra- to hexaploid *S. hegetschweileri* Heer em. Buser belong to the *S. phylicifolia* group sensu Rechinger [28], which can be assigned to the section *Phylicifoliae*. All three species are medium-sized shrubs. *S. bicolor* shows a disjunct distribution, while *S. hegetschweileri* is endemic to the central Alps. One stand of *S. bicolor* from the Alps is triploid [59]. For the other locations, neither chromosome counts nor genome size estimation exist to date. *Salix bicolor* has been the subject of controversial taxonomic treatments (see Table 1). Skvortsov [20] separated the Iberian species *S. basaltica* Coste from *S. bicolor*, and considered that *S. cantabrica* Rech.f., a tetraploid species from the Cordillera Cantabrica [57], should be included there. For *S. hegetschweileri*, chromosome counts of 4x, 5x, and 6x were reported, sometimes within one stand [58,59]. *Salix hegetschweileri* was separated by [28,54,68] and described as a variable, but clearly distinct species of the central Alps. However, during identification, confusion with glabrous forms of *S. myrsinifolia* is possible (Appendix A). Hybridization between both species can happen in overlapping distribution areas and was reported several times [54]. More problematic is the discrimination of *S. bicolor* from *S. phylicifolia*, which was subject to controversial discussions in recent decades. While both species were accepted by several authors [28,69,70], Skvortsov [20] treated the plants from the mountains of Central Europe not as distinct species but as subsp. of *S. phylicifolia (ssp. rhaetica*). Hexaploid *S. phylicifolia* L. *s.str.* is distributed in Northern Europe and Western Siberia.

### 4.2. Molecular Analyses

The leaves of a total of 64 newly sampled individuals, including material from type locations for *S. bicolor*, *S. hegetschweileri*, and *S. kaptarae*, were dried in silica gel and herbarium voucher specimens were deposited in the herbarium of the University of Goettingen (GOET). The sampling was supplemented by already published data to a final dataset of 145 samples (45 species). Detailed information for the sampling is summarized in Appendix A.

The DNA of all samples was extracted using the Qiagen DNeasy Plant Mini Kit following the manufacturer’s instructions (Valencia, CA, USA). After the quality check, the DNA was sent to Floragenex, Inc. (Portland, OR, USA) where the sequencing library preparation was conducted after [71] using the restriction enzyme *Pst*I (see [29] for details). Polyploids require an increased depth of coverage based on the larger genome size and the higher number of alleles [72,73]. Thus, we sequenced polyploid taxa on a separate plate to avoid loss of coverage. The quality of the resulting single-end 100bp long sequence reads was checked using FastQC v.0.10.1 [74]. After de-multiplexing, the reads were used to run ipyrad v.0.9.52 [75] with a clustering threshold of 85% and a minimum depth of eight reads for base calling. The clustering was performed with VSEARCH as implemented in ipyrad v.0.9.52. The maximum number of SNPs per locus was set to 20, and the maximum number of indels to 8. We set a threshold of a maximum of four alleles per site in the final cluster filtering. Filtering settings were optimized as described in [29]. Eventually, the m15 dataset (loci shared by at least 15 individuals) was used for phylogenetic analyses of the backbone topology. Additionally, we performed ipyrad for all samples of the sections *Nigricantes* and *Phylicifoliae* with a reduced backbone sampling to test for within-section relationships. Finally, it was conducted for each “hexaploid clade” separately applying the same settings.

We inferred phylogenetic relationships on concatenated alignments of the complete dataset as well as the reduced backbone dataset by using the GTR+ Γ model of nucleotide substitution implemented in RAXML v.8.2.4 [76]. We conducted, for each ML analysis, a rapid bootstrapping (BS) analysis with 100 replicates. In addition to BS, we applied quartet sampling (QS; [77]) with default settings to test the statistical support of a given topology. We ran 300 replicates using the –L option (minimum likelihood differential). QS is able to distinguish between conflicting signals and poor phylogenetic information. For each phylogeny shown in this paper, the observed QS values (QC/QD/QI) were visualized along with the BS values above and below branches.

### 4.3. Co-Ancestry Analysis and Genetic Structure

To explore the genetic structure of the hexaploid species, we employed RADpainter included in the package fineRADstructure [39], which infers population structure from RAD sequencing data. The program creates a co-ancestry similarity matrix based on haplotypes. The analysis compares nearest neighbor haplotypes by finding the closest relative for each allele for a given sample set using SNP data. For more details, see [39]. We performed the analyses using default settings based on the ‘alleles.loci’ file resulting from clade-specific ipyrad runs. First, we prepared the input file using the Python script fnRADstructure_input.py included in ‘finRADstructure-tools’ (https://github.com/edgardomortiz/fineRADstructure-tools (accessed on 8 January 2023)). With the input file, we calculated the co-ancestry matrix employing RADpainter. We then used ‘finestructure’ for clustering and tree assembly using 100,000 MCMC replicates and a burn-in of 100,000 applying the clustering approach ‘-m T’. The results were visualized with the R script ‘fineRADstructurePlot’.

Moreover, we performed genetic structure analyses applying SNMF within the R package LEA v.3.0.0 [78], which handles mixed-ploidy datasets, does not rely on assumptions of Hardy–Weinberg equilibrium, and is thus particularly suitable for analyzing polyploid apomicts [40,78,79]. We used .ustr files (one SNP per locus) generated by ipyrad and set the number of genetic clusters (K) from 1 to 10, maximal ploidy to 6, and repetitions to 8. To choose the number of ancestral K values, we used the implemented cross-entropy criterion. We retained and plotted multiple K values as the cross-entropy criterion only increased with increasing K values. The best run for each K value, i.e., the run with the lowest cross-entropy criterion, was selected to create bar plots in MS Excel.

### 4.4. Leaf Spectroscopy and Spectral Analyses

To measure the leaf spectra, we exclusively used well-dried herbarium material. Water might strongly affect the measurements and, by using herbarium samples, we could avoid any natural bias caused by different water contents. The sampling included 21 samples representing the six target species that were also used for molecular treatment supplemented by 17 additional samples to increase the sample size per species (Appendix A), totaling 38 samples from GOET. The reflectance measurements were conducted with a PSR+ portable spectroradiometer (Spectral Evolution, Haverhill, MA, USA) with bifurcated fiber optic cable and leaf probe clip. An “empty” white reflectance measurement was performed for calibration before each round of measurements. The reflectance spectra scanned from 350 to 2500 nm wavelength. We measured three times upper (adaxial) and three times lower (abaxial) surface to capture within-sample variability and statistical bias for each sample resulting in six scans per specimen. The measurement plots (spectra) were collected in the software DARWin SP Application Software v1.3 (Spectral Evolution).

The reflectance data were processed with a customized R script following [33]. In the first step, the data were filtered to remove spectra that resulted from erroneous scans with measured reflectance over 100%. Next, an average of all scans per sample was generated. The mean spectra per species were visualized as graphs. The results were forwarded to partial least squares discriminant analyses (PLS-DA), a multivariate analysis that classifies observation from PLS regression on indicator variables and proved to work well with high-dimensional multicollinear datasets as spectral data [31,80]. A prediction table was calculated based on the PLS scores. Then, we investigated the dataset with a principal component analysis (PCA). For this analysis, we defined different groups: “species level” (assignment of measurements to species determination) and “individual level” (assignment of spectra for each individual). The results were visualized with scatter plots with convex hulls or 95% confidence ellipses and with density plots.

### 4.5. Flow Cytometry

To analyze the ploidy levels of *S. bicolor* and *S. kaptarae*, flow cytometry (FC) was conducted using silica-gel-dried leaf material. Samples (~1 cm^2^) were reduced to small pieces using Tissue Lyzer II (Qiagen; 30 Hz, time 3 s). A total of 200 µL of 1% PVP Otto I extraction buffer were added, and samples were gently inverted for 1 min. Samples were filtered through CellTrics ^®^ filter (30 µm mesh, Sysmex Partec GmbH, Görlitz, Germany) into a flow cytometry sample tube. A total of 800 µL of DAPI-containing Otto II buffer [81] were then added to strain the DNA. FC analyses were carried out on a CyFlow Ploidy Analyzer (Sysmex, Norderstedt, Germany) and the Software CUBE16 v.1.6 (Sysmex, Norderstedt, Germany) was used to analyze the results and compute DNA content. Gain was set at 530 nm for an optimal differentiation between DNA content. Three different diploid *S. caprea* samples were used as external standard.

### 4.6. Morphometric Analyses

We conducted morphometric analyses on herbarium material to test for differences of *S. phylicifolia*, *S. bicolor*, and *S. hegetschweileri*. We examined the voucher specimens of all individuals used for molecular analysis, plus additional specimens from the same population, and additional populations for distinctive morphological characters as given in the literature (Table 1). Altogether 40 specimens were included. The original *S. bicolor* type population at Brocken, Harz Mountain, is well documented in the herbarium of the University of Göttingen (GOET) with several male and female individuals collected in the 18th and 19th century, including collections from Ehrhard. One isotype specimen of Ehrhard from the herbarium of the University of Vienna (WU) can also be included. The population has been almost destroyed due to constructions of buildings at the top of Brocken in the 20th century, but the remaining two individuals outside and cultivated in the Brocken garden that we used also for molecular and flow cytometric analyses derived from the original population [82]. The locus classicus of *S. hegetschweileri* in Urserental in the Alps of Switzerland is still occupied by a large healthy population, and both recent and old specimens could be used. *S. phylicifolia* L. was described from northern Sweden without indication of locality; we included specimens from this region. For all localities, see Appendix A.

From each specimen being at the stage of fruit maturity, we selected 5–8 fully developed leaves from the middle parts of branches to cover intra-individual variation. We measured leaf (blade) length, leaf (blade) width, angle at the blade’s base, length of longest teeth at leaf margin, and length of stipules. Length/width ratios of leaves were calculated. To estimate the density of indumentum, a character regarded as diagnostic for *S. bicolor* by [83], the hairs at lower leaf surface, were counted in a square of 5 × 5 mm under 10× magnification. However, in *S. bicolor* and in *S. phylicifolia*, indumentum density ceases continuously during the season, whereas *S. hegetschweileri* is glabrous. Therefore, the presence of hairs turned out to be extremely variable even between the leaves of one individual, and 87% of *S. bicolor* individuals also had glabrous leaves. Mean values were calculated for all measures for each individual, and these data were subjected to further statistical analyses using PAST vs. 4.03 [84]. Boxplots were created for the three species, and each variable was tested for normality using Shapiro–Wilk tests; since five of the six characters (except angle of blade base) were non-normally distributed, we calculated one-way ANOVAs, with Kruskal–Wallis tests and pairwise Mann–Whitney post hoc tests for statistical differences (for the angle of blade base, we additionally calculated Tukey tests that provided congruent results). Pairwise differences at the 0.05 significance level are indicated on boxplots with different letters; for *p*-values, see Supplement Appendix A. Data for density of hairs showed strong deviations from normality and failure of one-way ANOVAs because of a lack of variance. Because of its liability, the indumentum was excluded from further statistical analyses. Since length–width ratios revealed no significant differences between the three groups, this character was also excluded from multivariate statistics. With the remaining five characters, principal component analysis (PCA) and principal coordinate analysis (PCoA) using Gower’s distances were calculated, and scatter plots were produced with the respective first two axes. We visualized groups with different colors and convex hulls for three species and two species (*S. bicolor* + *S. phylicifolia* versus *S. hegetschweileri*). Discriminant analyses (DA) were calculated for the respective three and two predefined groups, and the percentages of correctly classified samples were calculated with confusion matrices.

## Figures and Tables

**Figure 1 plants-12-01144-f001:**
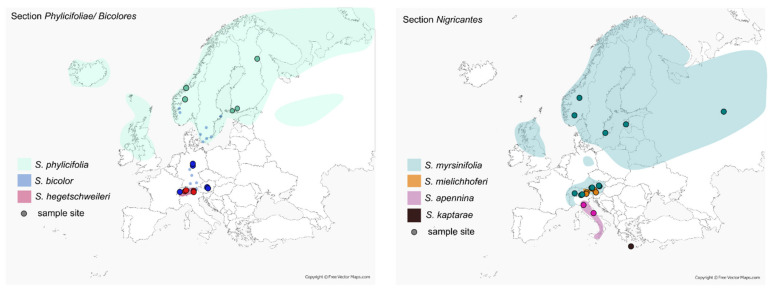
Distribution maps of the included hexaploid species of the sections *Nigricantes* and *Phylicifoliae*. Sample sites of the included accessions indicated as circles. Color coding according to legend within each map.

**Figure 2 plants-12-01144-f002:**
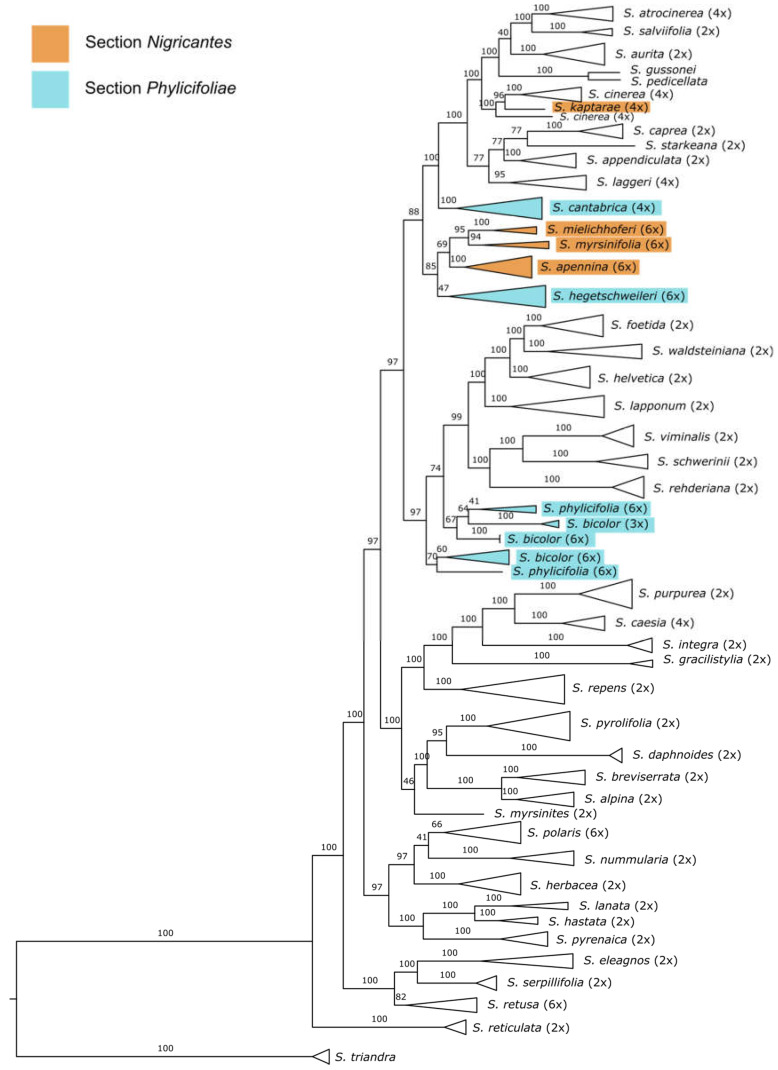
Backbone topology of the *Chamaetia*/*Vetrix* clade including 145 samples representing 45 Eurasian species plus *S. triandra* as outgroup based on a RAxML analyses on 46,568 RAD loci (630,764 SNPs) in a concatenated alignment of 5,341,180 bp and 66.95% missing sites. Bootstrap support values above branches. Ploidy level of species behind the species names. Species of the sections *Nigricantes* and *Phylicifoliae* are displayed in colors according to the legend. (Additional quartet sampling scores are displayed in Appendix A).

**Figure 3 plants-12-01144-f003:**
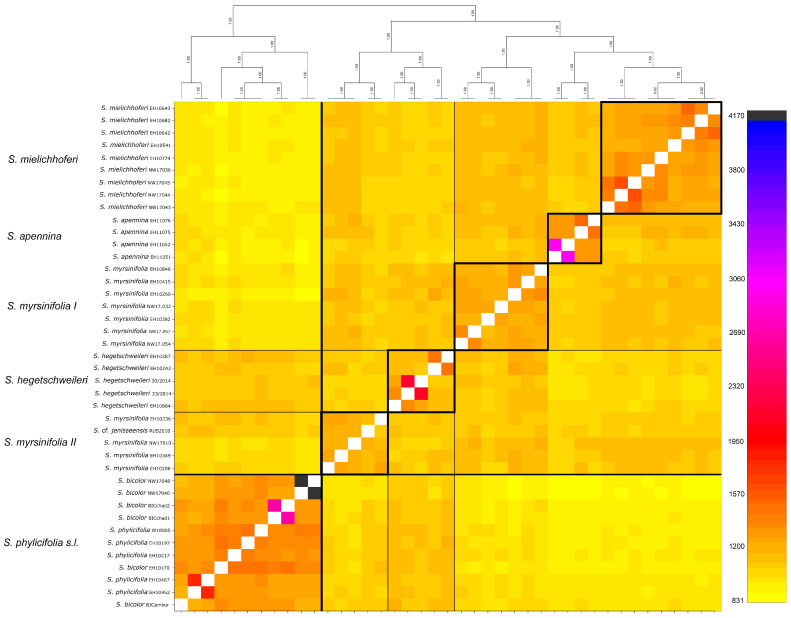
Co-ancestry analysis with fineRADstructure of 43 samples belonging to the sections *Nigricantes* and *Phylicifoliae* s.l. based on 49,754 RAD loci shared by at least 10 individuals. The amount of co-ancestry is illustrated by the color shade (legend left side). The sections and species are clearly separated. *Salix myrsinifolia* appears in two groups, one including samples from the European Alps (*S. myrsinifolia I*) and the other group includes samples from Fennoscandia and *S.* cf. *jenisseensis* from Russia (*S. myrsinifolia II*). *Salix bicolor* is included in *S. phylicifolia*. Detailed clade-wise results are presented in Appendix A.

**Figure 4 plants-12-01144-f004:**
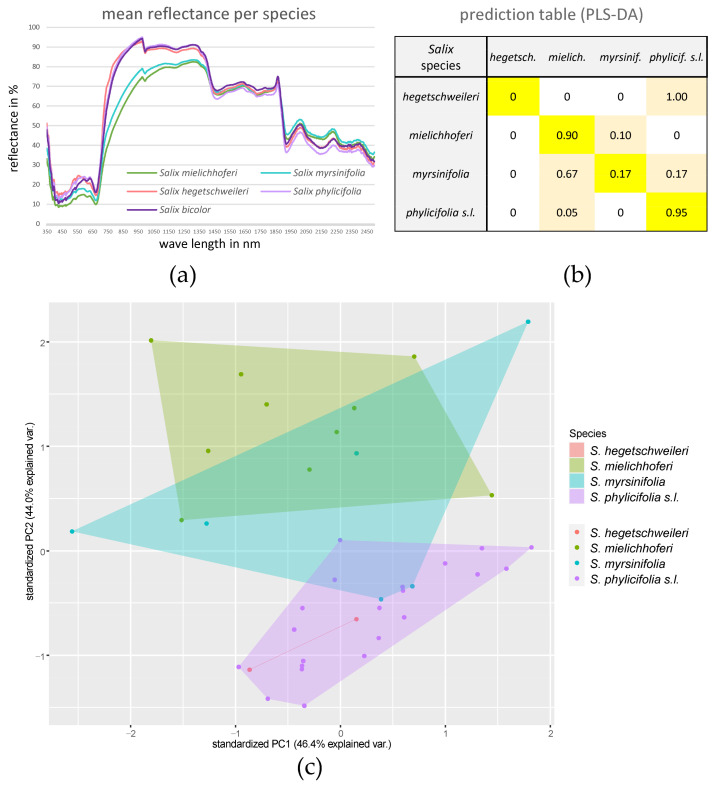
Infrared spectroscopy data. (**a**) Mean reflectance per species; species color-coded according to legend. (**b**) PLS-DA prediction table on spectral data showing relative numbers (percentage) of prediction. (**c**) PCA of spectral data analyses (convex hulls) of hexaploid species. The individual samples are color-coded according to the legend on the right. The displayed outcomes combine *S. bicolor* and *S. phylicifolia*.

**Figure 5 plants-12-01144-f005:**
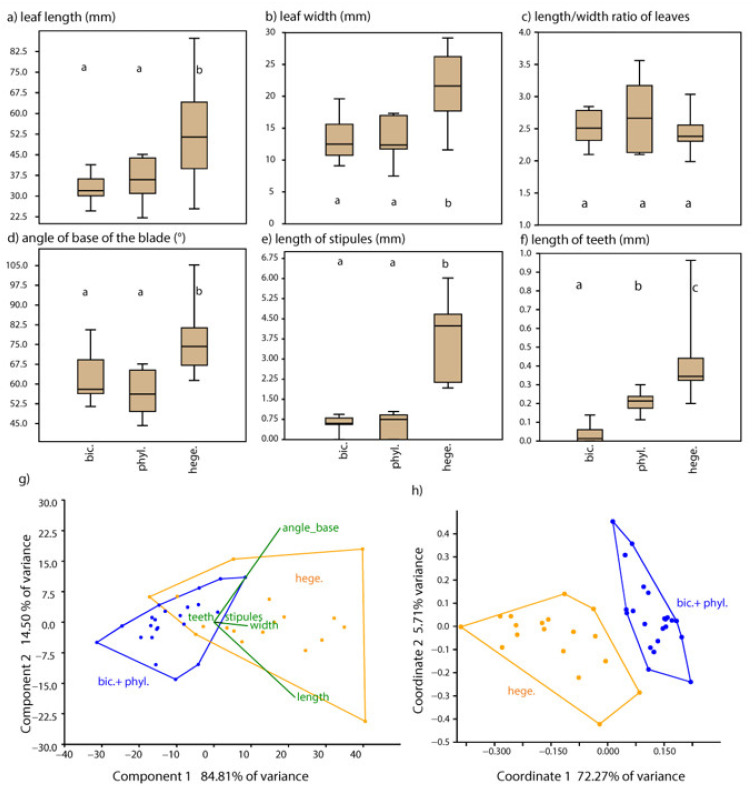
Morphometric analyses. (**a**–**f**) Boxplots of six characters plotted for *S. bicolor* (n = 15, bic.), *S. phylicifolia* (n = 7, phyl.), and *S. hegetschweileri* (n = 18, hege.). Significant pairwise differences according to Mann–Whitney tests are shown with different letters ((**a**,**b**); see Appendix A for all *p*-values). Boxes show the median and the quartile length, and whiskers the range. (**g**,**h**) Ordinations using the characters above except for (**c**); symbols and convex hulls according to the best grouping in discriminant analyses (97.5% correctly classified with two groups: *S. bicolor* and *S. phylicifolia* in blue, and *S. hegetschweileri* in orange). (**g**) PCA and (**h**) PCoA of five morphological traits.

**Table 1 plants-12-01144-t001:** Taxonomic treatments of polyploid willows of sect. *Nigricantes* and sect. *Phylicifoliae* by different authors and our results. See the Materials and Methods Section for more details.

Skvortsov (1999) [20]		Rechinger (1964) [28]; Aeschimann et al. (2004) [37]		Results of This Study	Distribution
*Subsect. Bicolores*		*S. phylicifolia* Group		*Sect. Phylicifoliae*	
*S. phylicifolia*	*subsp. phylicifolia*	*S. phylicifolia*	N. Europe, Russia	*S. phylicifolia*	N. Eurasia
*S. phylicifolia*	*subsp. rhaetica* (*incl. S. bicolor*)	*S. hegetschweileri*	Alps	(*incl. S. bicolor, excl. S. hegetschweileri*)	and Central European Mts.
*S. basaltica* (*? incl. S. cantabrica*)		*S. bicolor* (incl. *S. basaltica*)	European mts.	not analyzed	Not analyzed Pyrenees, Massif Central
* **Sect. Nigricantes** *		* **S. myrsinifolia group** *		* **Sect. Nigricantes** *	
				*S. hegetschweileri*	Alps (endemic)
*S. myrsinifolia*	*subsp. myrsinifolia*	*S. myrsinifolia*	Eurasia	*S. myrsinifolia s.l.*	Eurasia
*S. myrsinifolia*	*subsp. borealis*	*S. borealis*	N. Europe	*S. myrsinifolia s.l.*	Eurasia
*S. mielichhoferi*		*S. mielichhoferi*	Alps	*S. mielichhoferi*	Alps (endemic)
*S. apennina*		*S. apennina*	Apennines, S. Alps	*S. apennina*	Apennines, S. Alps
				*Sect. Vetrix*	
*-*		*S. cantabrica*	Iberian Peninsula	*S. cantabrica*	Iberian Peninsula (endemic)
*-*				*S. kaptarae* (*syn. of S. cinerea?*)	Crete

## Data Availability

The raw reads of molecular analysis can be found at NCBI Genbank SRA; Bioproject Accession Number PRJNA433286.

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
