# Peer review of "Insights into the Taxonomically Challenging Hexaploid Alpine Shrub Willows of Salix Sections Phylicifoliae and Nigricantes (Salicaceae)"

_plants, 2023, doi:10.3390/plants12051144_

Round 1

Reviewer 1 Report

This is a highly enlightening, analytical and interesting study of a very difficult genus. Easy to read and well written.

Line 10: medium-sized

Line 16: S. bicolor which are

Line 30: homologous

Line 32: which facilitates the colonization

Line 43: because of the reticulate

Line 49: The Chamaetia/Vetrix clade comprises about three-quarters

Line 53 and elsewhere:  medium-sized

Lines 75-76: by specific reflectance patterns

Line 111: bear phylogenetic signals

Line 129: comprised 66.95 % of missing data.

Line 132 and elsewhere: was in a sister position

Line 163: as a paraphyletic group

Line 192: showed a high number of

Line 232: explains 54.1% of the variation in the dataset.

Line 283: questions on the taxonomic treatment

Line 310: We, therefore, recommend excluding

Line 317: non-destructive,

Line 325: much fewer measurements

Line 330:  in the genus Myrcia

Line 347: of species of the previous

Lines 352-353: Besides gradual changes during seasonal development, phenotypic plasticity depending on habitat conditions causes large individual variations.

Line 392: the big ice shields

Line 411: due to the geographical separation

Line 422: confirm the proximity

Line 424: Instead, our data indicate the inclusion

Line 445: was not the focus

Line 450:  that show high similarity between

Line 492: based on shared ancestry

Line 510: Due to the lack of more samples, the

Line 527: was the subject of

Line 537: which was subject to controversial discussions

Line 549: After the quality check

Line 559: We set a threshold of a maximum of four alleles

Line 569: to test the statistical support

Line 653: from the middle parts

Line 656: To estimate the density

Line 659: Therefore, the presence

Author Response

Answers to Reviewer 1

We thank the reviewer for the helpful comments to improve our manuscript. We applied the recommended edits to our text.

Line 10: medium-sized  - done

Line 16: S. bicolor which are - done

Line 30: homologous – in case of allopolyploids the alleles of an allopolyploid individual originating from two different parental species are called “homeologs”. Therefore we kept “homeologous genes”

Line 32: which facilitates the colonization - done

Line 43: because of the reticulate - done

Line 49: The Chamaetia/Vetrix clade comprises about three-quarters - done

Line 53 and elsewhere:  medium-sized – done, changed in the entire manuscript.

Lines 75-76: by specific reflectance patterns - done

Line 111: bear phylogenetic signals - done

Line 129: comprised 66.95 % of missing data. - done

Line 132 and elsewhere: was in a sister position – we used the term “was in sister position” since it is commonly used to describe phylogenetic relationships.

Line 163: as a paraphyletic group. - done

Line 192: showed a high number of - done

Line 232: explains 54.1% of the variation in the dataset. - done

Line 283: questions on the taxonomic treatment - done

Line 310: We, therefore, recommend excluding - done

Line 317: non-destructive, - done

Line 325: much fewer measurements - done

Line 330:  in the genus Myrcia - done

Line 347: of species of the previous - done

Lines 352-353: Besides gradual changes during seasonal development, phenotypic plasticity depending on habitat conditions causes large individual variations. - done

Line 392: the big ice shields - done

Line 411: due to the geographical separation - done

Line 422: confirm the proximity - done

Line 424: Instead, our data indicate the inclusion - done

Line 445: was not the focus - done

Line 450:  that show high similarity between - done

Line 492: based on shared ancestry - done

Line 510: Due to the lack of more samples, the - done

Line 527: was the subject of - done

Line 537: which was subject to controversial discussions - done

Line 549: After the quality check - done

Line 559: We set a threshold of a maximum of four alleles - done

Line 569: to test the statistical support - done

Line 653: from the middle parts - done

Line 656: To estimate the density - done

Line 659: Therefore, the presence - done

Reviewer 2 Report

In this study, the authors use RAD-seq, infrared-spectroscopy, and morphometric data to solve allopolyploid Salix. It is an interesting manuscript for me and I don’t have the major comments. In the RAD-seq, I am worried about the ratio of missing data. For the ipyard pipeline of 145 samples, it comprised 66.95 % missing data. I know it always happens when the sample number is large. The phylogeny maybe not be the major part of this manuscript, however, it still affects the pattern of the phylogeny. The reference I put below discusses the missing data problem:

Deren A. R. Eaton, Elizabeth L. Spriggs, Brian Park, Michael J. Donoghue, Misconceptions on Missing Data in RAD-seq Phylogenetics with a Deep-scale Example from Flowering Plants, Systematic Biology, Volume 66, Issue 3, May 2017, Pages 399–412, https://doi.org/10.1093/sysbio/syw092

In my suggestion, the authors can find the prior phylogenetic tree that is constructed by other molecular data and compare it.

For section Nigricantes, the authors subset 29 samples of S. mielichhoferi, S. myrsinifolia, S. apennina and S. hegetschweileri to analysis and the missing data is low. It means the quality of SNPs is perfect and it is an important part of this manuscript. Therefore, in this manuscript, the phylogenetic tree of 45 species in 145 samples is only for support. If it is consistent with previous studies, it is advisable to add more evidence from previous studies to support the results of RAD-seq.

Author Response

We thank Reviewer 2 for the useful comments on our manuscript. We are aware of the influence of missing data to phylogenetic analyses. As mentioned by the reviewer, Eaton et al amongst others showed that missing data do not hamper the reconstruction of a phylogeny on a specific taxonomic level, whenever enough SNP information remains. We tested different thresholds in our parameter set beforehand, and decided finally for the settings mentioned in the manuscript that balance missing data and informative SNPs. Despite the high amount of missing data, the number of SNPs seems to be sufficient to resolve the phylogenetic tree. It is not possible to apply the same settings as for the subset of “Nigricantes” to the backbone phylogeny. On the higher taxonomic level including 145 samples, the observed locus dropout is much higher than in a closely related clade. Only few loci were shared by this many samples. Therefore, the amount of missing data might be low; however, the number of remaining loci will be also small and too low to allow for a resolved phylogenetic reconstruction. To address the comment we added some sentences about “missing data” in our discussion (Line 313 ff) and included the citation of “Eaton et al 2017” in the discussion to display the awareness of missing data in phylogenetic reconstructions.

Unfortunately, no phylogenetic tree based on other datasets exists that includes as many shrub willow species (Chamaetia/Vetrix clade) as our RAD phylogenies. We would have the chance to compare it to a non-resolved plastome phylogeny (Wagner et al 2021). A recently published preprint based on target capture (Sanderson et al 2023) includes several species of Salix Section Vetrix, but mainly North American species that cannot directly be compared. Additionally, no polyploids were included. Other existing trees were based on single markers and comprise much fewer samples. Thus, we are not able to supply the requested comparison with other datasets.